# Channel Gating Neural Networks

**Weizhe Hua**
wh399@cornell.edu

**Yuan Zhou**
yz882@cornell.edu

**Christopher De Sa**
cdesa@cornell.edu

**Zhiru Zhang**
zhiruz@cornell.edu

**G. Edward Suh**
gs272@cornell.edu

## Abstract

This paper introduces channel gating, a dynamic, fine-grained, and hardware-efficient pruning scheme to reduce the computation cost for convolutional neural networks (CNNs). Channel gating identifies regions in the features that contribute less to the classification result, and skips the computation on a subset of the input channels for these ineffective regions. Unlike static network pruning, channel gating optimizes CNN inference at run-time by exploiting input-specific characteristics, which allows substantially reducing the compute cost with almost no accuracy loss. We experimentally show that applying channel gating in state-of-the-art networks achieves 2.7-8.0× reduction in floating-point operations (FLOPs) and 2.0-4.4× reduction in off-chip memory accesses with a minimal accuracy loss on CIFAR-10. Combining our method with knowledge distillation reduces the compute cost of ResNet-18 by 2.6× without accuracy drop on ImageNet. We further demonstrate that channel gating can be realized in hardware efficiently. Our approach exhibits sparsity patterns that are well-suited to dense systolic arrays with minimal additional hardware. We have designed an accelerator for channel gating networks, which can be implemented using either FPGAs or ASICs. Running a quantized ResNet-18 model for ImageNet, our accelerator achieves an encouraging speedup of 2.4× on average, with a theoretical FLOP reduction of 2.8×.

## 1 Introduction

The past half-decade has seen unprecedented growth in the use of machine learning with convolutional neural networks (CNNs). CNNs represent the state-of-the-art in large scale computer vision, natural language processing, and data mining tasks. However, CNNs have substantial computation and memory requirements, greatly limiting their deployment in constrained mobile and embedded devices [1, 23]. There are many lines of work on reducing CNN inference costs, including low-precision quantization [3, 21], efficient architectures [14, 30], and static pruning [9, 12, 19, 22]. However, most of these techniques optimize a network *statically*, and are agnostic of the input data at run time. Several recent efforts propose to use additional fully-connected layers [6, 8, 28] or recurrent networks [20, 29] to predict if a fraction of the computation can be skipped based on certain intermediate results produced by the CNN at run time. These approaches typically perform coarse-grained pruning where an entire output channel or layer is skipped dynamically.

In this paper, we propose *channel gating*, a dynamic, fine-grained, and hardware-efficient pruning scheme, which exploits the spatial structure of features to reduce CNN computation at the granularity of individual output activation. As illustrated in Figure 1, the essential idea is to divide a CNN layer into a *base* path and a *conditional* path. For each output activation, the base path obtains a partial sum of the output activation by performing convolution on a subset of input channels. The activation-wise gate function then predicts whether the output activations are effective given the partial sum. Only the effective activations take the conditional path which continues computing on the rest of the input channels. For example, an output activation is *ineffective* if the activation is clipped to zero by ReLU

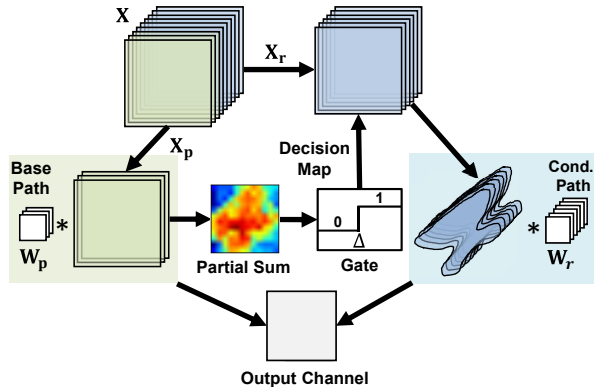

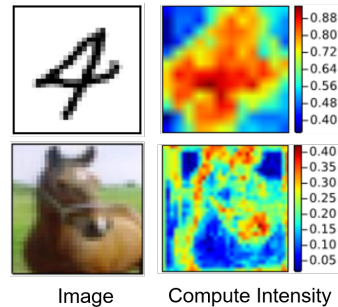

Figure 1: Illustration of *channel gating* — A subset of input channels (colored in green) are used to generate a decision map, and prune away unnecessary computation in the rest of input channels (colored in blue).

Figure 2: Computation intensity map — The computation intensity map is obtained by averaging decision maps over output channels.

as it does not contribute to the classification result. Our empirical study suggests that the partial and final sums are strongly correlated, which allows the gate function to make accurate predictions on whether the outputs are effective. Figure 2 further conceptualizes our idea by showing the heat maps of the normalized computation cost for classifying two images, with the "cool" colors indicating the computation that can substantially be pruned by channel gating.

Clearly, the channel gating policy must be learned through training, as it is practically infeasible to manually identify the "right" gating thresholds for all output channels without causing notable accuracy loss. To this end, we propose an effective method to train CNNs with channel gating (**CGNets**) from scratch in a single pass. The objective is to maximize the reduction of computational cost while minimizing the accuracy loss. In addition, we introduce channel grouping to ensure that all input channels are included and selected equally without a bias for generating dynamic pruning decisions. The experimental results show that channel gating achieves a higher reduction in floating-point operations (FLOPs) than existing pruning methods with the same or better accuracy. In addition to pruning computation, channel gating can be extended with a coarser-grained channel-wise gate to further improve memory efficiency by reducing the off-chip memory accesses for weights.

CGNet is also hardware friendly for two key reasons: (1) channel gating requires minimal additional hardware (e.g., small comparators for gating functions) to support fine-grained pruning decisions on a dense systolic array; (2) channel gating maintains the locality and the regularity in both computations and data accesses, and can be efficiently implemented with small changes to a systolic array architecture similar to the Google TPU [16]. Our ASIC accelerator design achieves a 2.4× speed-up for the CGNet that has the theoretical FLOP reduction of 2.8×.

This paper makes the following major contributions:

- We introduce *channel gating*, a new lightweight dynamic pruning scheme, which is trainable and can substantially improve the computation and memory efficiency of a CNN at inference time. In addition, we show that channel grouping can naturally be integrated with our scheme to effectively minimize the accuracy loss. For ImageNet, our scheme achieves a higher FLOP reduction than state-of-the-art pruning approaches [5, 8, 11, 18, 22, 31] with the same or less accuracy drop.
- Channel gating represents a new layer type that can be applied to many CNN architectures. In particular, we experimentally demonstrate the benefits of our technique on ResNet, VGG, binarized VGG, and MobileNet models.
- We co-design a specialized hardware accelerator to show that channel gating can be efficiently realized using a dense systolic array architecture. Our ASIC accelerator design achieves a 2.4× speed-up for the CGNet that has the theoretical FLOP reduction of 2.8×.

## 2  Related Work

**Static Pruning.** Many recent proposals suggest pruning unimportant filters/features statically [12, 19, 22, 31]. They identify ineffective channels in filters/features by examining the magnitude of the

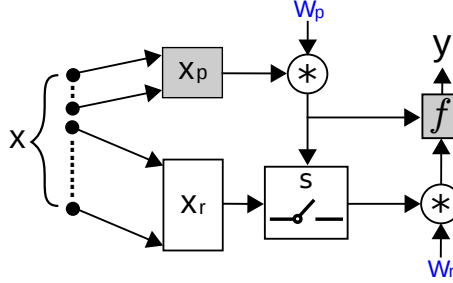
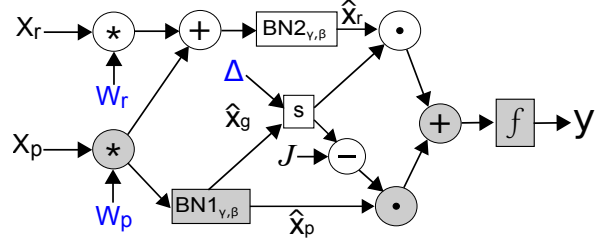

Figure 3: Channel gating block — $\mathbf{x_p}, \mathbf{W_p}$ and $\mathbf{x_r}, \mathbf{W_r}$ are the input features and weights to the base and conditional path, respectively. $s$ is the gate function which generates a binary pruning decision based on the partial sum for each output activation. $f$ is the activation function.

Figure 4: The computational graph of channel gating for training — $\Delta$ is a per-output-channel learnable threshold. $\mathbf{J}$ is an all-one tensor of rank three. Subtract the binary decision $d$ from $J$ gives the complimentary of decision which is used to select the activations from the base path.

weights/activation in each channel. The relatively ineffective subset of the channels are then pruned from the model. The pruned model is then retrained to mitigate the accuracy loss from pruning. By pruning and retraining iteratively, these approaches can compress the model size and reduce the computation cost. PerforatedCNN [7] speeds up the inference by skipping the computations of output activations at fixed spatial locations. Channel gating can provide a better trade-off between accuracy and computation cost compared to static approaches by utilizing run-time information to identify unimportant receptive fields in input features and skipping the channels for those regions dynamically.

**Dynamic Pruning.** Figurnov et al. [6] introduce the spatially adaptive computation time (SACT) technique on Residual Network [10], which adjusts the number of residual units for different regions of input features. SACT stops computing on the spatial regions that reach a predefined confidence level, and thus only preserving low-level features in these regions. McGill and Perona [24] propose to leverage features at different levels by taking input samples at different scales. Instead of bypassing residual unit, dynamic channel pruning [8] generates decisions to skip the computation for a subset of output channels. Teja Mullapudi et al. [28] propose to replace the last residual block with a Mixture-of-Experts (MoE) layer and reduce computation cost by only activating a subset of experts in the MoE layer. The aforementioned methods embed fully-connected layers in a baseline network to help making run-time decisions. Lin et al. [20] and Wu et al. [29] propose to train a policy network with reinforcement learning to make run-time decisions to skip computations at channel and residual block levels, respectively. Both approaches require additional weights and extra FLOPs for computing the decision. In comparison, CGNet generates fine-grained decisions with no extra weights or computations. Moreover, instead of dropping unimportant features, channel gating approximates these features with their partial sums which can be viewed as cost-efficient features. Our results show that both fine-grained pruning and high-quality decision functions are essential to significantly reduce the amount of computation with minimal accuracy loss.

## 3 Channel Gating

In this section, we first describe the basic mechanism of channel gating. Then, we discuss how to design the gate function for different activation functions. Last, we address the biased weight update problem by introducing channel grouping.

### 3.1 Channel Gating Block

Without loss of generality, we assume that features are rank-3 tensors consisting of $c$ channels and each channel is a 2-D feature of width $w$ and height $h$. Let $\mathbf{x}_l, \mathbf{y}_l, \mathbf{W}_l$ be the input features, output features, and weights of layer $l$, respectively, where $\mathbf{x}_l \in \mathbb{R}^{c_l \times w_l \times h_l}$, $\mathbf{y}_l \in \mathbb{R}^{c_{l+1} \times w_{l+1} \times h_{l+1}}$, and $\mathbf{W}_l \in \mathbb{R}^{c_{l+1} \times c_l \times k_l \times k_l}$. A typical block of a CNN layer includes convolution ($*$), batch normalization (BN) [15], and an activation function ($f$). The output feature can be written as $\mathbf{y}_l = f(\text{BN}(\mathbf{W}_l * \mathbf{x}_l))^1$.

To apply channel gating, we first split the input features statically along the channel dimension into two tensors where $\mathbf{x}_l = [\mathbf{x_p}, \mathbf{x_r}]$. For $\eta \in (0,1]$, $\mathbf{x_p}$ consists of $\eta$ fraction of the input channels while the rest of the channels form $\mathbf{x_r}$, where $\mathbf{x_p} \in \mathbb{R}^{\eta c_l \times w_l \times h_l}$ and $\mathbf{x_r} \in \mathbb{R}^{(1-\eta)c_l \times w_l \times h_l}$. Similarly, let $\mathbf{W_p}$ and $\mathbf{W_r}$ be the weights associated with $\mathbf{x_p}$ and $\mathbf{x_r}$. This decomposition means that $\mathbf{W}_l * \mathbf{x}_l = \mathbf{W_p} * \mathbf{x_p} + \mathbf{W_r} * \mathbf{x_r}$. Then, the partial sum $\mathbf{W_p} * \mathbf{x_p}$ is fed into the gate to generate a binary decision tensor ($\mathbf{d} \in \{0,1\}^{c_{l+1} \times w_{l+1} \times h_{l+1}}$), where $\mathbf{d}_{i,j,k} = 0$ means skipping the computation on the rest of the input channels (i.e., $\mathbf{x_r}$) for $\mathbf{y}_{i,j,k}$.

Figure 3 illustrates the structure of the channel gating block for inference[2]. There exist two possible paths with different frequency of execution. We refer to the path which is always taken as the *base path* (colored in grey) and the other path as the *conditional path* given that it may be gated for some activations. The final output ($\mathbf{y}$ in Figure 3) is the element-wise combination of the outputs from both the base and conditional paths. The output of the channel gating block can be written as follows, where $s$ denotes the gate function and $i, j, k$ are the indices of a component in a tensor of rank three:

$$\mathbf{y}_{l\,i,j,k} = \begin{cases} f(\mathbf{W_p} * \mathbf{x_p})_{i,j,k}, & \text{if } \mathbf{d}_{i,j,k} = \mathrm{s}(\mathbf{W_p} * \mathbf{x_p})_{i,j,k} = 0 \\ f(\mathbf{W_p} * \mathbf{x_p} + \mathbf{W_r} * \mathbf{x_r})_{i,j,k}, & \text{otherwise} \end{cases} \tag{1}$$

Channel gating works only if the partial sum is a good predictor for the final sum. We hypothesize that the partial sum is strongly correlated with the final sum. We test our hypothesis by measuring the linear correlation between the partial and final sums of a layer with different $\eta$ value. The average Pearson correlation coefficient of 20 convolutional layers in ResNet-18 over 1000 training samples equals 0.56, 0.72, and 0.86 when $\eta$ is $\frac{1}{8}$, $\frac{1}{4}$, and $\frac{1}{2}$, respectively. The results suggest that the partial and final sums are still moderately correlated even when only $\frac{1}{8}$ of the channels are used to compute the partial sum. While partial sums cannot accurately predict the exact values for all output activations, we find that they can effectively identify and approximate ineffective output activations.

## 3.2 Learnable Gate Functions

To minimize the computational cost, the gate function should only allow a small fraction of the output activations to take the conditional path. We introduce a per-output-channel learnable threshold ($\Delta \in \mathbb{R}^{c_l}$) to learn different gating policies for each output channel and define the gate function using the Heaviside step function, where $i, j, k$ are the indices of a component in a tensor of rank three.

$$\theta(\mathbf{x})_{i,j,k} = \begin{cases} 1, & \text{if } \mathbf{x}_{i,j,k} \geq 0 \\ 0, & \text{otherwise} \end{cases} \tag{2}$$

An output activation is considered to be ineffective if the activation is zeroed out by ReLU or saturated to the limit values by sigmoid or hyperbolic tangent. Thus, the gate function is designed based on the activation function in the original network. For instance, if ReLU is used, we use $\mathrm{s}(\mathbf{x}, \Delta) = \theta(\mathbf{x} - \Delta)$ as the gate function. Similarly, if the activation function has limit values such as hyperbolic tangent and sigmoid function, we apply $\mathrm{s}(\mathbf{x}, \Delta_h, \Delta_l) = \theta(\Delta_h - \mathbf{x}) \circ \theta(\mathbf{x} - \Delta_l)$ as the gate function where $\circ$ is the Hadamard product operator and $\Delta_h, \Delta_l$ are the upper and lower thresholds, respectively. The same gate can be applied to binary neural networks [3, 26]. The rationales for choosing the gate function are twofold: (1) the step function is much cheaper to implement in hardware compared to the Softmax and Noisy Top-K gates proposed in [27]; (2) the gate function should turn off the rest of the channels for activations which are likely to be ineffective.

We use the gate for ReLU as a more concrete example. Let $\tau$ be the fraction of activations taking the conditional path. To find $\Delta$ which satisfies $P(\mathbf{W_p} * \mathbf{x_p} \geq \Delta) = \tau$, we normalize the partial sums using batch normalization without scaling and shift. During inference, the batch normalization is merged with the gate to eliminate extra parameters and computation. The merged gate is defined as:

$$\widetilde{\mathrm{s}}(\mathbf{x}, \boldsymbol{\Delta}) = \theta\left(\mathbf{x} - \Delta \cdot \sqrt{\mathrm{Var}(x)} - \mathbf{E}[x]\right) \tag{3}$$

where $\mathbf{E}[x]$ and $\mathrm{Var}(x)$ are the mean and variance, respectively. The merged gate has $c_{l+1}$ thresholds and performs $w_{l+1} \cdot h_{l+1} \cdot c_{l+1}$ point-wise comparisons between the partial sums and the thresholds.

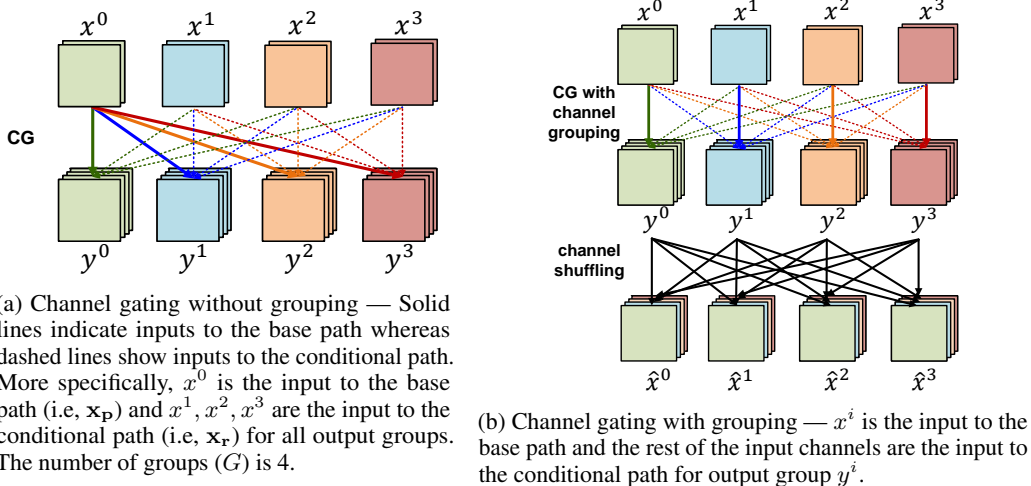

(a) Channel gating without grouping — Solid lines indicate inputs to the base path whereas dashed lines show inputs to the conditional path. More specifically, $x^0$ is the input to the base path (i.e, $\mathbf{x_p}$) and $x^1, x^2, x^3$ are the input to the conditional path (i.e, $\mathbf{x_r}$) for all output groups. The number of groups ($G$) is 4.

(b) Channel gating with grouping — $x^i$ is the input to the base path and the rest of the input channels are the input to the conditional path for output group $y^i$.

Figure 5: Illustration of channel gating with and without channel grouping.

In addition to reducing computation, we can extend channel gating to further reduce memory footprint. If the conditional path of an entire output channel is skipped, the corresponding weights do not need to be loaded. For weight access reduction, we introduce a per-layer threshold ($\tau_c \in \mathbb{R}^l$). Channel gating skips the entire output channel if less than $\tau_c$ fraction of activations are taking the conditional path. Here the channel-wise gate function generates a binary pruning decision for each output channel which is defined as:

$$\mathbf{S}(\mathbf{x}, \Delta, \tau_c) = \theta(\sum_{j,k} s(\mathbf{x}, \Delta) - \tau_c \cdot w_{l+1} \cdot h_{l+1})_i \qquad (4)$$

where $i$, $j$, and $k$ are the indices of channel, width, and height dimensions, respectively. The channel-wise gate adds one more threshold and $c_{l+1}$ additional comparisons. Overall, channel gating remains a lightweight pruning method, as it only introduces $(w_{l+1} \cdot h_{l+1} + 1) \cdot c_{l+1}$ comparisons with $c_{l+1} + 1$ thresholds per layer.

## 3.3 Unbiased Channel Selection with Channel Grouping

In previous sections, we assume a predetermined partition of input channels between the base path ($\mathbf{x_p}$) and the conditional path ($\mathbf{x_r}$). In other words, a fixed set of input channels is used as $\mathbf{x_p}$ for each layer. Since the base path is always taken, the weights associated with $\mathbf{x_p}$ (i.e., $\mathbf{W_p}$) will be updated more frequently than the rest during training, making the weight update process biased. Our empirical results suggest that such biased updates can cause a notable accuracy drop. Thus, properly assigning input channels to the base and conditional paths without a bias is critical in minimizing the loss.

We address the aforementioned problem with the help of channel grouping. Inspired partly by grouped convolution, channel grouping first divides the input and output features into the same number of groups along the channel dimension. Let $\mathbf{x}_l^i$, $\mathbf{y}_l^i$, $\mathbf{W}_l^i$ be the $i$-th group of input features, output features, and weights in a channel gating block, respectively, where $\mathbf{x}_l^i \in \mathbb{R}^{\eta c_l \times w_l \times h_l}$, $\mathbf{y}_l^i \in \mathbb{R}^{\eta c_{l+1} \times w_{l+1} \times h_{l+1}}$, and $\mathbf{W}_l^i \in \mathbb{R}^{\eta c_{l+1} \times c_l \times k_l \times k_l}$. Then, for the $i$-th output group, we choose the $i$-th input group as $\mathbf{x_p}$ and rest of the input groups as $\mathbf{x_r}$. Let $\mathbf{x_p}^i$ and $\mathbf{x_r}^i$ be $\mathbf{x_p}$ and $\mathbf{x_r}$ for the $i$-th output group, respectively. Channel gating with $G$ groups can be defined by substituting $\mathbf{x_p}^i = \mathbf{x}_l^i$ and $\mathbf{x_r}^i = [\mathbf{x}_l^0, ..., \mathbf{x}_l^{i-1}, \mathbf{x}_l^{i+1}, ..., \mathbf{x}_l^{G-1}]$ into Equation (1). The number of groups ($G$) is set to be $\frac{1}{\eta}$ as each input group should contain the $\eta$ fraction of all input channels. Intuitively, the base path of CGNet is an ordinary grouped convolution.

Figure 5b shows an example that illustrates channel grouping. In Figure 5a, we do not apply channel grouping and $x^0$ is always used for the base path for all output groups. In Figure 5b where channel grouping is applied, $x^0$ is fed to the base path only for output group $y^0$. For other output groups, $x^0$ is used for the conditional path instead. In this way, we can achieve unbiased channel selection where every input channel is chosen as the input to the base path once and conditional path ($G - 1$)

times (here $G = 4$). As a result, all weights are updated with the same frequency without a bias. To improve cross-group information flow, we can further add an optional channel shuffle operation, as proposed in ShuffleNet [30]. Figure 5b illustrates the shuffling using the black solid lines between the output groups of the current layer and the input groups of the next layer.

## 4 Training CGNet

We leverage the gradient descent algorithm to train CGNet. Figure 4 illustrates the computational graph during training. Applying batch normalization directly on the output diminishes the contribution from the base path since the magnitude of $\mathbf{W_p} * \mathbf{x_p}$ can be relatively small compared to $\mathbf{W_p} * \mathbf{x_p} + \mathbf{W_r} * \mathbf{x_r}$. To balance the magnitude of the two paths, we apply two separate batch normalization (BN 1 and BN 2) before combining the outputs from the two paths[3]. We subtract the output of the gate from an all-one tensor of rank three ($\mathbf{J} \in \mathbb{R}^{c_{l+1} \times w_{l+1} \times h_{l+1}}$) to express the if condition and combine the two cases with an addition which makes all the operators differentiable except the gate function.

In addition, we show two important techniques to make CGNet end-to-end learnable and reduce the computation cost: **(1)** approximating a non-differentiable gate function; **(2)** inducing sparsity in decision maps during training. Last, we also discuss boosting accuracy of CGNet using knowledge distillation.

**Approximating non-differentiable gate function.** As shown in Figure 4, we implement a custom operator in MxNet [2] which takes the outputs ($\hat{\mathbf{x}}_r$, $\hat{\mathbf{x}}_p$, $\hat{\mathbf{x}}_g$) from the batch normalization as the inputs and combines the two paths to generate the final output ($\mathbf{y}$). The gradients towards $\hat{\mathbf{x}}_r$ and $\hat{\mathbf{x}}_p$ are rather straightforward whereas the gradient towards $\hat{\mathbf{x}}_g$ and $\Delta$ cannot be computed directly since the gate is a non-differentiable function. We approximate the gate with a smooth function which is differentiable with respect to $\mathbf{x}$ and $\Delta$. Here, we propose to use $\mathbf{s}(\mathbf{x}, \Delta) = \frac{1}{1+e^{\epsilon \cdot (\mathbf{x} - \Delta)}}$ to approximate the gate during backward propagation when the ReLU activation is used. $\epsilon$ is a hyperparameter which can be tuned to adjust the difference between the approximated function and the gate. With the approximated function, the gradients $d\hat{\mathbf{x}}_g$ and $d\Delta$ can be calculated as follows:

$$d\hat{\mathbf{x}}_g = -d\Delta = d\mathbf{y} \cdot (\hat{\mathbf{x}}_p - \hat{\mathbf{x}}_r) \cdot \frac{-\epsilon \cdot e^{\epsilon(\hat{\mathbf{x}}_g - \Delta)}}{\mathbf{s}(\hat{\mathbf{x}}_g, \Delta)^2} \tag{5}$$

**Inducing sparsity.** Without loss of generality, we assume that the ReLU activation is used and propose two approaches to reduce the FLOPs. As the input ($\hat{\mathbf{x}}_g$) to the gate follows the standard normal distribution, the pruning ratio (F) increases monotonically with $\Delta$. As a result, reducing the computation cost is equivalent to having a larger $\Delta$. We set a target threshold value named target ($T$) and add the squared loss of the difference between $\Delta$ and $T$ into the loss function.

We also show that this proposed approach works better than an alternative approach empirically in terms of trading off the accuracy for FLOP reduction. The alternative approach directly optimizes the objective by adding the computation cost as a squared loss term (computation-cost loss = $(\sum_l (\sum_c \sum_w \sum_h (\mathbf{J} - \mathbf{s}(\mathbf{W_p} * \mathbf{x_p}))) \cdot \eta c_l \cdot k_l^2 \cdot w_{l+1} \cdot h_{l+1} \cdot c_{l+1})^2)$. We observe that adding the computation-cost loss prunes layers with higher FLOPs and introduces an imbalanced pruning ratio among the layers while the proposed approach keeps a more balanced pruning ratio. As a result, we add the $\lambda \sum_l (T - \Delta_l)^2$ term to the loss function, where $\lambda$ is a scaling factor.

**Knowledge distillation (KD).** KD [13] is a model compression technique which trains a student network using the softened output from a teacher network. Let T, S, $\mathbf{y_t}$, $\mathbf{P_T^\kappa}$, $\mathbf{P_S^\kappa}$ represent a teacher network, a student network, ground truth labels, and the probabilistic distributions of the teacher and student networks after softmax with temperature ($\kappa$). The loss of the student model is as follows:

$$\mathcal{L}_S(W) = -((1 - \lambda_{kd}) \sum \mathbf{y_t} \log(\mathbf{P_S^\kappa}) + \lambda_{kd} \sum \mathbf{P_T^\kappa} \log(\mathbf{P_S^\kappa})) \tag{6}$$

As a result, the student model is expected to achieve the same level of prediction accuracy as the teacher model. We leverage KD to improve the accuracy of CGNets on ImageNet where a ResNet-50 model is used as the teacher of our ResNet-18 based CGNets with $\kappa = 1$ and $\lambda_{kd} = 0.5$.

Table 1: The accuracy and the FLOP reduction of CGNets for CIFAR-10 without KD.

| Baseline | # of Groups | Target Threshold | Top-1 Error Baseline (%) | Top-1 Error Pruned (%) | Top-1 Accu. Drop (%) | FLOP Reduction |
|---|---|---|---|---|---|---|
| ResNet-18 | 8 | 2.0 | 5.40 | 5.44 | 0.04 | 5.49× |
| | 16 | 3.0 | 5.40 | 5.96 | 0.56 | 7.95× |
| Binary VGG-11 | 8 | 1.0 | 16.85 | 16.95 | 0.10 | 3.02× |
| | 8 | 1.5 | 16.85 | 17.10 | 0.25 | 3.80× |
| VGG-16 | 8 | 1.0 | 7.20 | 7.12 | -0.08 | 3.41× |
| | 8 | 2.0 | 7.20 | 7.59 | 0.39 | 5.10× |
| MobileNetV1 | 8 | 1.0 | 12.15 | 12.44 | 0.29 | 2.88× |
| | 8 | 2.0 | 12.15 | 12.80 | 0.65 | 3.80× |

# 5 Experiments

We first evaluate CGNets only with the activation-wise gate on CIFAR-10 [17] and ImageNet (ILSVRC 2012) [4] datasets to compare the accuracy and FLOP reduction trade-off with prior arts. We apply channel gating on a modified ResNet-18[4], binarized VGG-11, VGG-16, and MobileNetV1 on CIFAR-10. For ImageNet, we use the ResNet-18, ResNet-34, and MobileNetV1 as the baseline. Furthermore, we explore channel gating with activation-wise and channel-wise gates to reduce both computation cost and off-chip memory accesses. We choose a uniform target threshold ($T$) and number of groups ($G$) for all CGNets for the experiments in Section 5.1 and 5.2. Last, we show that the accuracy and FLOP reduction trade-off of CGNets can be further improved by exploring the design space.

## 5.1 Reduction in Computation Cost (FLOPs)

In Table 1, we show the trade-off between accuracy and FLOP reduction when CGNet models are used for CIFAR-10 without KD. Channel gating can trade-off accuracy for FLOP reduction by varying the group size and the target threshold. CGNets reduce the computation by 2.7 - 8.0× with minimal accuracy degradation on five state-of-the-art architectures using the two gate functions proposed in Section 3.2. It is worth noting that channel gating achieves a 3× FLOP reduction with negligible accuracy drop even for a binary model (Binary VGG-11).

Table 2 compares our approach to prior arts [5, 8, 11, 18, 22, 31] on ResNet and MobileNet without KD. The results show that channel gating *outperforms* all alternative pruning techniques, offering smaller accuracy drop *and* higher FLOP saving. Discrimination-aware channel pruning [31] achieves the highest FLOP reduction among three static pruning approaches on ResNet-18. The top-1 accuracy drop of channel gating (CGNet-A) is **1.9%** less than discrimination-aware channel pruning, which demonstrates the advantage of dynamic pruning. Feature Boosting and Suppression (FBS) [8] is a channel-level dynamic pruning approach which achieves higher FLOP saving than static pruning approaches. Channel gating is much simpler than FBS, yet achieves **1.6%** less accuracy drop and slightly higher FLOP reduction (CGNet-B). Channel gating also works well on a lightweight CNN built for mobile and embedded application such as MobileNet. MobileNet with channel gating (CGNet-A) achieves **1.2%** higher accuracy with larger FLOP saving than a thinner MobileNet model (0.75 MobileNet). We believe that channel gating outperforms existing pruning approaches for two reasons: (1) instead of dropping the ineffective features, channel gating approximates the features with the partial sums; 2) channel gating performs more fine-grained activation-level pruning.

Table 3 shows additional comparisons between CGNet with and without channel grouping and knowledge distillation (KD) on ResNet-18 for ImageNet. CGNet with channel grouping achieves 0.9% higher top-1 accuracy and 20% higher FLOP reduction than the counterpart without channel grouping. Applying KD further boosts the top-1 accuracy of CGNet by 1.3% and improves the FLOP saving from 1.93× to 2.55×. We observe the same trend for CIFAR-10 where channel grouping improves the top-1 accuracy by 0.8% when the computation is reduced by 5×. KD does not improve the model accuracy on CIFAR-10. The small difference between the ground truth label and the output from the teacher model makes the distilled loss ineffective.

Table 2: Comparisons of accuracy drop and FLOP reduction of the pruned models for ImageNet without KD — CGNet A and B represent CGNet models with different target thresholds and scaling factors.

| Baseline | Model | Dynamic | Top-1 Error Baseline (%) | Top-1 Error Pruned (%) | Top-1 Accu. Drop (%) | FLOP Reduction |
|---|---|---|---|---|---|---|
| ResNet-18 | Soft Filter Pruning [11] | ✗ | 29.7 | 32.9 | 3.2 | 1.72× |
| | Network Slimming [22] | ✗ | 31.0 | 32.8 | 1.8 | 1.39× |
| | Discrimination-aware Pruning [31] | ✗ | 30.4 | 32.7 | 2.3 | 1.85× |
| | Low-cost Collaborative Layers [5] | ✓ | 30.0 | 33.7 | 3.7 | 1.53× |
| | Feature Boosting and Suppression [8] | ✓ | 29.3 | 31.8 | 2.5 | 1.98× |
| | **CGNet-A** | ✓ | 30.8 | 31.2 | **0.4** | **1.93×** |
| | **CGNet-B** | ✓ | 30.8 | 31.7 | **0.9** | **2.03×** |
| ResNet-34 | Filter Pruning [18] | ✗ | 26.8 | 27.9 | 1.1 | 1.32× |
| | Soft Filter Pruning [11] | ✗ | 26.1 | 28.3 | 2.1 | 1.70× |
| | **CGNet-A** | ✓ | 27.6 | 28.7 | **1.1** | **2.02×** |
| | **CGNet-B** | ✓ | 27.6 | 29.8 | **2.2** | **3.14×** |
| Mobile-Net | 0.75 MobileNet [14] | ✗ | 31.2 | 33.0 | 1.8 | 1.75× |
| | **CGNet-A** | ✓ | 31.2 | 31.8 | **0.6** | **1.88×** |
| | **CGNet-B** | ✓ | 31.2 | 32.2 | **1.0** | **2.39×** |

Table 3: Comparisons of the CGNet-A with and without channel grouping and knowledge distillation on ResNet-18 for ImageNet.

| Channel Grouping | KD | Top-1 Error Pruned (%) | Top-1 Accu. Drop (%) | FLOP Reduction |
|---|---|---|---|---|
| ✗ | ✗ | 32.1 | 1.3 | 1.61× |
| ✓ | ✗ | 31.2 | 0.4 | 1.93× |
| ✓ | ✓ | 30.3 | **-0.5** | **2.55×** |
| ✓ | ✓ | 31.1 | **0.3** | **2.82×** |

Table 4: Power, performance, and energy comparison of different platforms — Batch size for ASIC and CPU/GPU is 1 and 32, respectively.

| Platform | Intel i7-7700k | NVIDIA GTX 1080Ti | ASIC Baseline | CGNet |
|---|---|---|---|---|
| Frequency (MHz) | 4200 | 1923 | 800 | 800 |
| Power (Watt) | 91 | 225 | **0.20** | 0.25 |
| Throughput (fps) | 13.8 | 1563.7 | 254.3 | **613.9** |
| Energy/Img (mJ) | 6594.2 | 143.9 | 0.79 | **0.41** |

## 5.2 Reduction in Memory Accesss for Weights

As discussed in Section 3.2, a channel-wise gate can be introduced into the channel gating block to reduce the memory accesses for weights. Figure 6 shows the accuracy, the reduction in weight accesses, and the FLOP reduction of CGNets with different target thresholds for the activation-wise gate ($T \in \{1.5, 2.0\}$) and the channel-wise gate ($\tau_c \in \{0, 0.05, 0.10, 0.20\}$). CGNets with the channel-wise gate reduces the number of weight accesses by **3.1×** and **4.4×** with **0.6%** and **0.3%** accuracy degradation when $T$ equals 1.5 and 2.0, respectively. We find that CGNets with a larger $T$ achieve larger reduction in weight accesses with less accuracy drop as fewer activations are taking the conditional path when $T$ becomes larger.

## 5.3 Design Space Exploration

CGNets with a uniform target threshold ($T$) and number of groups ($G$) only represent one design point among many possible CGNet architectures. We explore the design space of CGNets by varying the number of groups in each residual module. Specifically, we fix the group size of one specific module and then vary the group sizes of the other modules to obtain multiple design points per residual module. As depicted in Figure 7, each residual module may favor a different $G$. For example, for the second module, CGNets with two groups provide a better trade-off between accuracy and FLOP reduction compared to the ones with 16 groups. In contrast, for the fourth module, CGNets with 16 groups outperform the designs with two groups. Similarly, we can explore the design space by choosing a different $T$ for each residual module. This study shows that one can further improve the accuracy and FLOP reduction trade-off by searching the design space.

## 5.4 Speed-up Evaluation

The evaluation so far uses the FLOP reduction as a proxy for a speed-up. PerforatedCNN [7] introduces sparsity at the same granularity as CGNet, and shows that the speed-ups on CPUs/GPUs closely match the FLOP saving. Moreover, there is a recent development on a new GPU kernel called sampled dense matrix multiplications [25], which can potentially be leveraged to implement

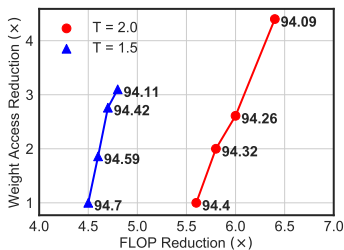

Figure 6: Weight access reduction on ResNet-18 for CIFAR-10 with different $T$ and $\tau_c$ combinations — Annotated values represent the top-1 accuracy.

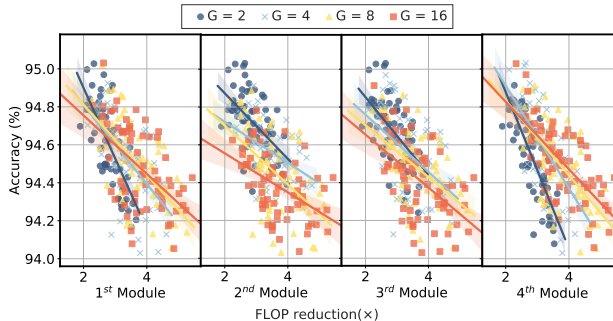

Figure 7: Comparisons of accuracy and FLOP reduction of CGNets with different group size for each residual module on ResNet-18 for CIFAR-10 — The lines represent the linear regression for the corresponding design points.

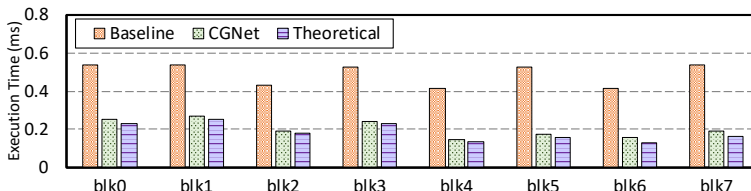

Figure 8: Execution time breakdown for each residual block — The theoretical execution time of CGNet represents the best possible performance under a specific resource usage, and is computed as the total number of multiplications (FLOPs) per inference divided by the number of multipliers.

the conditional path of CGNets efficiently. Therefore, we believe that the FLOP reduction will also translate to a promising speed-up on CPUs/GPUs.

To evaluate the performance improvement and energy saving of applying channel gating for specialized hardware, we implemented a hardware prototype targeting a TSMC 28nm standard cell library. The baseline accelerator adopts the systolic array architecture similar to the Google TPU. CGNet requires small changes to the baseline accelerator because it reuses partial sums and only need additional comparators to make gating decisions. In addition to the comparators, CGNet uses a custom data layout and memory banking to support sparse data movements after pruning.

Figure 8 shows the speed-up of CGNet over the baseline accelerator. We compare the FLOP reduction and the actual speed-up. The actual speed-up is $2.4\times$ when the FLOP reduction, which represents the theoretical speed-up, is $2.8\times$. This result shows that hardware can effectively exploit the dynamic sparsity in channel gating. Moreover, the small gap between the execution time of CGNet and the theoretical execution time suggests that CGNet is hardware-efficient. As shown in Table 4, the CGNet outperforms a CPU by $42.1\times$ in terms of throughput and is four orders of magnitude more energy-efficient. Compared with an NVIDIA GTX GPU, the CGNet is $326.3\times$ more energy-efficient.

# 6 Conclusions and Future Work

We introduce a dynamic pruning technique named channel gating along with a training method to effectively learn a gating policy from scratch. Experimental results show that channel gating provides better trade-offs between accuracy and computation cost compared to existing pruning techniques. Potential future work includes applying channel gating on objection detection tasks.

# 7 Acknowledgments

This work was partially sponsored by Semiconductor Research Corporation and DARPA, NSF Awards #1453378 and #1618275, a research gift from Xilinx, Inc., and a GPU donation from NVIDIA Corporation. The authors would like to thank the Batten Research Group, especially Christopher Torng (Cornell Univ.), for sharing their Modular VLSI Build System. The authors also thank Zhaoliang Zhang and Kaifeng Xu (Tsinghua Univ.) for the C++ implementation of channel gating and Ritchie Zhao and Oscar Castañeda (Cornell Univ.) for insightful discussions.

## Footnotes

[1] The bias term is ignored because of batch normalization. A fully-connected layer is a special case where $w$, $h$, and $k$ all are equal to 1.

[2]Batch normalization is omitted as it is only a scale and shift function during inference.

[3]BN 1 and BN 2 share the same parameters ($\gamma, \beta$).

[4]The ResNet-18 variant architecture is similar to the ResNet-18 for ImageNet while using $3 \times 3$ filter in the first convolutional layer and having ten outputs from the last fully-connected layer.

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
