[Supplementary Material]

# A Supplementary Material

## A.1 Set Hyperparameters

The proposed channel gating building block introduces four hyperparameters — $\epsilon$, $\eta$, $T$, and $\lambda$. $\epsilon$ is the hyperparameter in the approximated gate function. $\eta$ is the fraction of input channels in $\mathbf{x}_p$ of each layer. $T$ and $\lambda$ are the target threshold and scaling factor in the squared loss term which is used to reduce the FLOPs during training. We experimented with settings of these hyperparameters on CIFAR-10 with ResNet-18:

- $\epsilon$: The maximum value of the partial derivative of the gate with respect to $\mathbf{x_p}$ and $\Delta$ equals to $\frac{\epsilon}{4}$. To avoid gradient vanishing and explosion, we explore three different $\epsilon$ values ($\epsilon = 2, 3, 4$).
- $\eta$: We set a uniform $\eta$ for all the layers and examine the impact of different $\eta$ on the accuracy and FLOP reduction trade-off. Moreover, different layer might requires different number of channels ($\eta c_{l-1}$) in the base path to minimize the accuracy loss. However, it is impractical to search for the best $\eta$ for each layer which is a combinatorial search problem.
- $T$: Similar to $\eta$, a uniform $T$ is chosen for all the layers whereas different $T$ values are exploited to achieve different FLOP reduction.
- $\lambda$: The $\lambda$ determines whether the threshold can reach the target ($T$) and and how many epochs needed to reach the target. If the $\lambda$ is too large, it is equivalent to initialize the $\Delta$ with $T$. In this paper, we simply set the $\lambda$ to be relative small value ($10^{-4}$) to prevent drastic changes of the $\Delta$ value.

(a) Accuracy vs. FLOP reduction with different $\epsilon$.    (b) Accuracy vs. FLOP reduction with different $\eta$.

Figure 7: Empirical study on choosing hyperparameters.

We first study the impact of the $\epsilon$ on the accuracy and FLOP reduction trade-off by sweeping three target values ($T = 0, 1, 2$) whereas keeping all other hyperparameters fixed ($\eta = \frac{1}{8}$, $\lambda = 10^{-4}$). Figure (7a) shows that the $\epsilon = 2$ provides a better trade-off empirically. For rest of the experiments, we use $\epsilon = 2$. We then compare the effectiveness of having different fractions of the input channels in the base path . Three $\eta$ values ($\eta = \frac{1}{4}, \frac{1}{8}, \frac{1}{16}$) are evaluated and the results are depicted in Figure (7b). Based on the experimental results, we find that a smaller $\eta$ provides better accuracy and FLOP reduction trade-off if targeting a lower FLOP saving.

## A.2 Datasets

The CIFAR-10 dataset consists of 50,000 training images and 10,000 test images corresponding to 10 classes. We use a standard data-augmentation scheme in [22], in which the images are zero-padded with four pixels on each side, randomly cropped to produce $32 \times 32$ images, and horizontally mirrored. For testing, we evaluate the model using the original $32 \times 32$ image. The ImageNet dataset contains a total of 1.2 million training images and 50,000 validation images with 1000 different classes. We follow the same data augmentation scheme adopt by [10]. The image is resized to $256 \times 480$ pixels randomly. We scale the image by $\frac{256}{480}$, subtract the per-pixel mean, and random flip the image horizontally. A $224 \times 224$ random crop is taken from an image. Moreover, the color augmentation in [18] is used. We use the $256 \times 256$ center cropped image to test the model.

## A.3 Training Details

**CIFAR-10:** We train all models using stochastic gradient descent (SGD) with a momentum weight of 0.9 and weight decay factor of $10^{-4}$. For CIFAR-10, all models are trained with mini-batch size 256 for 300 epochs. The initial learning rate is 0.1 and we divide the learning rate by 10 twice at epoch 200 and 250.

**ImageNet:** All models are trained using Nesterov accelerated stochastic gradient descent (SGD) with a momentum weight of 0.9 and weight decay factor of $10^{-4}$. All models are trained with mini-batch size 256 for 140 epochs. The initial learning rate is 0.1 and divided by 10 at epoch 30, 60, 90, and 120.

## A.4 Ablation Study

The core idea of *CGNet* lies in channel gating, grouping, and shuffle operations. To evaluate the benefits of introducing channel grouping and shuffle, we compare the accuracy and FLOP reduction between *CGNet*s with and without the grouping and shuffle operations on CIFAR-10. As shown in

Figure 8: Accuracy vs. FLOP reduction with and without the channel grouping and shuffle operations.

Figure 8, integrating CGNet with channel grouping improves the test accuracy significantly. Channel grouping guarantees unbiased weight updates. The top-1 accuracy of CGNet with channel grouping is 0.84% higher than the one without channel grouping when the computational cost is reduced by $5\times$.

Moreover, as discussed in Section 3.3, channel shuffle operation enhances the cross-channel information flow when pruning the computations aggressively. CGNet with channel shuffle outperforms the one without channel shuffle when achieving more than $3\times$ FLOP reduction. The top-1 accuracy of CGNet with channel shuffle is 0.23% higher than the one without channel shuffle when targeting at $5\times$ FLOP reduction.

## A.5 Empirical Study of Gate Functions

| Gate Functions | Computational Overhead | | Test Error (%) | FLOP Reduction |
|---|---|---|---|---|
| | $+$ | $\times$ | | |
| $\ell^1$-norm | $c_{l-1} \cdot w_{l-1} \cdot h_{l-1}$ | 0 | 8.19 | $3.97\times$ |
| FC | $c_{l-1} \cdot w_{l-1} \cdot h_{l-1}$ | $c_l \cdot c_{l-1}$ | 6.50 | $4.25\times$ |
| $CG_{\text{coarse}}$ | $c_l \cdot w_{l-1} \cdot h_{l-1}$ | 0 | 6.40 | $4.15\times$ |
| $CG_{\text{fine}}$ | **0** | **0** | **5.44** | **$5.49\times$** |

Table 5: Comparison between different gate functions on CIFAR-10.

There exist two categories of gate functions in literature — parameter-less and parameterized gates. A parameter-less gate predicts the saliency of features based on the statistics without extra parameters whereas a parameterized gate function usually embeds a fully-connected layer to generate the saliency vectors. In contrast, our gating scheme generates activation-wise saliency at nearly zero cost by reusing a fraction of the weights and computations within a layer. It is also feasible to combine methods for generating saliency vectors proposed in previous studies [6, 8, 29] with the learnable Heaviside step function. In this subsection, we empirically compare channel gating and the two previously-proposed gating functions.

Let $\mathbf{x_l}$ be the input features of layer $l$ where $\mathbf{x_l} \in \mathbb{R}^{c_{l-1} \times w_{l-1} \times h_{l-1}}$. $\ell^1$-norm gate reduces $\mathbf{x_l}$ to a saliency vector ($\mathbf{s}_{\ell^1-\mathbf{norm}} \in \mathbb{R}^{c_{l-1}}$) by calculating the $\ell^1$-norm of each input channel. FC gate further applies a fully-connected layer with $c_l$ neurons on $\mathbf{s}_{\ell^1-\mathbf{norm}}$ to obtain the saliency vector $\mathbf{s}_{FC}$ $\in \mathbb{R}^{c_l}$ for output channels. $CG_{\text{coarse}}$ and $CG_{\text{fine}}$ represent the proposed gating scheme with different granularity of the decision. $CG_{\text{fine}}$ generates decision for each output activation as discussed in Section 3 whereas $CG_{\text{coarse}}$ produces a single decision for an entire output channel by aggregating the fine-grained decisions.

As shown in Table 5, the FC gate outperforms $\ell^1$-norm gate because it introduces the sparsity in both input and output channels, providing a quadratic reduction in computational cost with respect to the pruning ratio. $CG_{\text{coarse}}$ achieves a similar trade-off between accuracy and FLOPs compared to the FC gate, but with lower computation overhead because it re-uses partial sums to predict the saliency of output activations. Finally, the proposed $CG_{\text{fine}}$ provides the highest accuracy and FLOP savings with no overhead. Although it is feasible to generate activation-wise decisions using the FC gate, the computational overhead of the FC gate grows exponentially with the width/height of 2-D features as it requires having $c_l \cdot w_l \cdot h_l$ neurons.

## A.6 Qualitative Results

Figure 9: Samples with different FLOP reductions for CIFAR-10.

In this subsection, we present more qualitative results of CGNet. In Figure 9, We also show the test samples with the maximum and minimum FLOP reduction for five categories in CIFAR-10. There exists more than $2\times$ difference in FLOP reduction among these samples which demonstrates that *CGNet* can prune adaptively for different samples.