[Reviews · NeurIPS 2019]

Reviewer 1



Originality: The proposed idea is not very different from other dynamic pruning methods. In my opinion the main contribution is the reduced amount of extra computation needed for the pruning that allows interesting computational gains and the GPU friendly way of pruning based on channels. The use of channel grouping to avoid a biased selection of the channels is also quite interesting. In dynamic pruning authors should also cite: [Convolutional Networks with Adaptive Inference Graphs. Andreas Veit Serge Belongie]. Quality: The proposed contribution makes sense and is justified by interesting experiments on CIFAR10 and Imagenet. Clarity: Overall the paper is well written and not difficult to read. Thanks to fig. 3 and 4 it should not be too difficult to reproduce the model. Significance: Conditional computation (in this case dynamic pruning), although in principle very interesting, often does not deliver really effective approaches. Often the theoretical speed-up is already reduced by the fact that for choosing which part of the network to compute it requires additional computation. Additionally, the theoretical gain is then difficult to convert to a real speed-up due to the constraints on the data of GPU architectures. The proposed approach is one of the first to show that conditional computation can lead to real speed-ups by a very light control architecture and GPU friendly pruning. This is the most important contribution of this work. Additional comments: - Why Knowledge distillation is applied only on ImageNet? - In speed-up evaluation the authors mention that the algorithm should be fast on GPU as for Perforated CNN. However, they do not really evaluate the method speed-up on GPU. - I think the method should be compared to also other approaches for speeding up convolutions, even if not based on pruning such as Perforated Convolutions. Final Evaluation: I read the other reviews and authors feedback. Authors answers were clear and I did not find any important issue that I overlooked. Thus, I confirm my evaluation.

Reviewer 2



Pruning in CNN models has gained a lot of attention in the recent years and this paper introduces a new dynamic channel pruning technique. The paper introduces a simple and effective dynamic channel pruning technique along with accelerators for ASIC hardware showing actual execution time speedups. Pros 1. The paper does a good job covering the related work for pruning in CNN models. Static vsndynamic pruning and channel vs parameter pruning are well explained. The channel gating layer proposed by the authors is dynamic and more fine-grained than [8] which is the closest related paper to this work. However, references and work related to sparsity in the parameters is missing. 2. The channel gating layer introduces a gating mechanism designed on the activation function. The channel block is simple and effective. It doesn't add much more compute on top of the existing layer and requires to compute partial sums all of which seem current hardware friendly. 3. The experimental results are significant across various models and datasets. The authors show an improved FLOP reduction while getting better Accuracies than other pruning methods referenced in the paper. Also, using knowledge distillation the FLOP reduction gains are furthered. 4. Real time execution speeds ups on ASIC hardware. The speedup is very close to the theoretical gains validating the results further. Cons 1. Dynamic channel pruning is explored in multiple related works. The differences seem small between the various techniques. 2. Dynamic pruning doesn't save storage space. It'll be interesting to compare the FLOP reduction of sparse (weight sparsity) models vs channel pruning models to understand the tradeoff between Accuracy and FLOP reduction further.

Reviewer 3



- Dynamic pruning idea is very interesting. Compared with static pruning ideas, dynamic pruning may use more parameters without increase the runtime during inference. - Authors realized the idea by using partial sum of a subset of input channels to generate a decision map. The description of training is clear in general, however, the motivation of using partial sum to make decision and why this helps with the results were not explained. Meanwhile, in the inference, given a test example, how to decide the base channels and the optional channels? - The paper is technically sound and fairly amount of experimental results were presented to demonstrate the effectiveness of the proposed idea.

[Author Response · NeurIPS 2019]

We thank all reviewers for the helpful comments. **Figure A** and **Table A** are newly added to address the questions
raised in the reviews. **Table 1, 2, 3** refer to the tables in the original paper.

**Common question on comparison to other CNN optimization techniques including parameter-sparse models:** In
Table A, we first compare CGNet with a non-pruning based approach on AlexNet. CGNet achieves 1.7% less top-5
accuracy drop and $1.3\times$ higher FLOP reduction compared to PerforatedCNN [7]. Channel gating works well with
other non-pruning optimization such as binarization (Binary Network in Table 1) and efficient architecture (MobileNets
in Table 1 and 2). CGNet also outperforms two weight sparse compression techniques. As the dense layers only
account for a small fraction of the overall computation cost, the FLOP reduction of sparse weight pruning is not as
impressive as the weight reduction. CGNet achieves 0.9% and 0.8% less accuracy drop and $1.8\times$ and $1.1\times$ higher
FLOP reduction than the models in Table A (second and third rows), respectively. Moreover, we believe that channel
gating is complementary to weight pruning approaches as channel gating exploits input-dependent feature sparsity.

Figure A: Distribution of channels used across layers.

| Model | Top-1 & Top-5 Error Baseline (%) | | Top-1 & Top-5 Accu. Drop (%) | | FLOP Reduction |
|---|---|---|---|---|---|
| M. Figurnov et al. (NIPS'16) [7] | / | 19.6 | / | 2.3 | $2.1\times$ |
| W. Wen et al. (NIPS'16) | 43.7 | / | 1.8 | / | $1.5\times$ |
| X. Zhu et al. (IJCAI'18) | 43.7 | / | 1.7 | / | $2.4\times$ |
| **CGNet** | 41.9 | 19.4 | **0.9** | **0.6** | **$2.7\times$** |

TABLE A: Comparisons of accuracy drop and FLOP reduction of the pruned models on AlexNet for ImageNet.

**Reviewer#1**
**Q1. Knowledge distillation (KD) on CIFAR-10:** KD does not improve the model accuracy on CIFAR-10. The small
difference between the ground truth label and the output from the teacher model makes the distilled loss ineffective.

**Q2. Speed-up on GPU:** Our current focus is to demonstrate channel gating on custom hardware, similar to Google
TPUs. It is worth noting that there is a recent development on a new GPU kernel called sampled dense matrix
multiplications, which can potentially be leveraged to implement the conditional path of CGNets efficiently.

**Reviewer#5**
**Q1. Storage size:** While channel gating does not reduce storage size, it can be extended to reduce off-chip memory
accesses of the customized accelerator by dynamically pruning the entire output channel. We performed a preliminary
study on ResNet-18 for CIFAR-10, and obtained a 46% reduction in off-chip memory accesses with 0.2% accuracy loss
when we pruned channels with 10% or less salient activations.

**Q2. An analysis of the distribution of channels used across layers:** In Figure A, we show the percentage of channels
used for each layer of ResNet-18 for CIFAR-10 (first row in Table 1). We observe that later layers use fewer channels
than earlier layers and $1\times1$ conv layers (layer 3, 8, 13, and 18) skip more channels than $3\times3$ conv layers.

**Reviewer#7**
**Q1. Rationales of using partial sum for gating decision:** The partial and final sums are strongly correlated, which
makes the partial sum a good estimator for the final output. The correlation coefficient is 0.86 when half of the channels
are used to compute the partial sum (line 124). The fact that existing pruning approaches can effectively prune input
channels and use a partial sum as the final output also suggests that the partial sum is a good approximate. Moreover,
unlike other dynamic pruning approaches that embed additional fully-connected layers or even RNNs to make decisions,
using the partial sum only requires minimal additional compute and hardware to support fine-grained pruning.

**Q2. Channel selection in the inference time:** We did not select channels manually. Both $\mathbf{x_p}$ and $\mathbf{x_r}$ are chosen
statically for both training and inference. The basic solution (first row in Table 3) simply uses the first $\chi$ fraction of
channels as $\mathbf{x_p}$. The channels in the base path which is always taken will be "favored" during training and naturally
become more important than those in the conditional path. Alternatively, we propose channel grouping and shuffling
to "equalize" the importance of each channel. Channel grouping divides the input and output features into the same
number of groups along the channel dimension. Then, for the $i$-th output group, we choose the $i$-th input group as
$\mathbf{x_p}$ and rest of the input groups as $\mathbf{x_r}$ statically, which makes base path an ordinary grouped convolution. As a result,
each channel is selected as $\mathbf{x_p}$ and $\mathbf{x_r}$ with the same frequency. Our empirical result shows that CGNet with channel
grouping achieves 0.9% higher top-1 accuracy and 20% higher FLOP reduction than the counterpart without grouping.

**Q3. Gate function with channel shuffling:** The gate function is an element-wise operation which is applied to the
partial sum. Channel shuffling operates on the final output after finishing the base and conditional paths. The two
functions operate on different inputs with no interference.

**Q4. Trained model and inference code:** We made a C++ implementation and pretrained models of both baseline and
CGNet inference available in an anonymous git repository. This implementation is unoptimized and only meant to
demonstrate the idea. We are also cleaning up the code of the ASIC implementation and will make it open source soon.

[Meta-Review · NeurIPS 2019]

The paper presents a simple yet effective way to reduce computation by only computing a sub-part of the inner-products. This idea results in realization speedups as confirmed by an ASIC design. Given the similarity with some existing work on dynamic pruning, I recommend acceptance as a poster.